Trichoderma based formulations control the wilt disease of chickpea (Cicer arietinum L.) caused by Fusarium oxysporum f. sp. ciceris, better when inoculated as consortia: findings from pot experiments under field conditions

Chohan Safeer A. 1
Akbar Muhammad muhammad.akbar@uog.edu.pk 1
Iqbal Umer 2 3
1 Department of Botany, University of Gujrat , Gujrat , Punjab , Pakistan
2 Crop Diseases Research Institute, National Agricultural Research Centre , Islamabad , Pakistan
3 Seed Health Lab., Plant Genetic Resources Institute, National Agricultural Research Centre , Islamabad , Pakistan
Mora-Montes Héctor
Electronic publication date: 2024 Aug 19
Publication date: 2024
Volume: 12
Electronic Location ID: e17835
Received 2024 Mar 7; Accepted 2024 Jul 9
Copyright: ©2024 Chohan et al.
Copyright year: 2024
Copyright holder: Chohan et al.
License: This is an open access article distributed under the terms of the Creative Commons Attribution License, which permits unrestricted use, distribution, reproduction and adaptation in any medium and for any purpose provided that it is properly attributed. For attribution, the original author(s), title, publication source (PeerJ) and either DOI or URL of the article must be cited.
License URL: https://creativecommons.org/licenses/by/4.0/

Keywords: Trichoderma, Fusarium oxysporum, Wilt, Disease, Chickpea, Cicer arietinum, Crop, Biological control

Funding: The authors received no funding for this work.

==============================
Background

Commercial/chemical pesticides are available to control Fusarium wilt of chickpea, but these antifungals have numerous environmental and human health hazards. Amongst various organic alternatives, use of antagonistic fungi like Trichoderma, is the most promising option. Although, Trichoderma spp. are known to control Fusarium wilt in chickpea but there are no reports that indicate the biocontrol efficacy of indigenous Trichoderma spp. against the local pathogen, in relation to environmental conditions.

Methods

In the present study, biological control activity of Trichoderma species formulations viz., Trichoderma asperellum, Trichoderma harzianum (strain 1), and Trichoderma harzianum (strain 2), either singly or in the form of consortia, was investigated against Fusarium oxysporum f. sp. ciceris, the cause of Fusarium wilt in chickpea, in multiyear pot trials under open field conditions. The antagonistic effect of Trichoderma spp. was first evaluated in in vitro dual culture experiments. Then the effects of Trichoderma as well as F. oxysporum, were investigated on the morphological parameters, disease incidence (DI), and disease severity (DS) of chickpea plants grown in pots.

Results

In dual culture experiments, all the Trichoderma species effectively reduced the mycelial growth of F. oxysporum. T. asperellum, T. harzianum (strain 1), and T. harzianum(strain 2) declined the mycelial growth of F. oxysporumby 37.6%, 40%, and 42%. In open field pot trials, the infestation of F. oxysporum in chickpea plants significantly reduced the morphological growth of chickpea. However, the application of T. asperellum, T. harzianum (strain 1), and T. harzianum (strain 2), either singly or in the form of consortia, significantly overcome the deleterious effects of the pathogen, thereby resulted in lower DI (22.2% and 11.1%) and DS (86% and 92%), and ultimately improved the shoot length, shoot fresh weight and shoot dry weight by 69% and 72%, 67% and 73%, 68% and 75%, during the years 1 and 2, respectively, in comparison with infested control. The present study concludes the usefulness and efficacy of Trichoderma species in controlling wilt disease of chickpea plants under variable weather conditions.

Introduction

Chickpea (Cicer arietinum L.) is an essential crop worldwide. It belongs to family fabaceae. Chickpea has annual production of over 10 million tons (Muehlbauer & Sarker, 2017). It is cultivated on over 13.5 million hectares and contributes 5.72% to total global production, annually (Jendoubi et al., 2017).

Factors responsible for less production of chickpeas include both biotic and abiotic stresses. Biotic factors include fungal, bacterial, and viral attacks (Roorkiwal et al., 2018). Numerous pathogens including fungi, nematodes, and viruses have been identified which lessen the chickpeas production. Out of these pathogens, Fusarium spp. significantly affect the chickpeas and cause different soil and airborne diseases (Rocha et al., 2023). Fusarium wilt of chickpea is the most devastating disease caused by fungal species named as Fusarium oxysporum, causing yield loss by 60% (Singh & Yadav, 2007). However, under favorable environmental conditions for the pathogen, yield losses have been estimated as high as 100% (Hashem, Tabassum & Abd_Allah, 2020).

Various strategies have been utilized to manage Fusarium wilt caused by F. oxysporum. Synthetic fungicides are widely used to control such devastating soil-borne pathogens. As an example, carbendazim treatment declined the Fusarium wilt by 24% and enhanced the yield of chickpea up to 28% (Khan et al., 2014). However, synthetic fungicides have their own limitations such as high cost, environmental pollution and adverse effects on soil microbiota, which play important roles in soil organic matter decomposition (Hoyos-Carvajal, Orduz & Bissett, 2009). Moreover, many phytopathogens have developed resistance against these fungicides as demonstrated in Fusarium verticillioides and F. oxysporum (Xu et al., 2019; Poromarto & Permatasari, 2023).

Biological control is considered an alternative method to manage the soil-borne pathogens (Biswas & Ali, 2017). Soil inhabiting filamentous fungus, Trichoderma species (Ascomycota) have been known for their capability as biocontrol agent (BCA) (Khan et al., 2014; Wonglom et al., 2019), against many plant pathogens (Khan et al., 2014; Hernández-Melchor, Ferrera-Cerrato & Alarcón, 2019). As an example, in a dual culture experiment, Trichoderma resulted in suppression of Fusarium colony growth by 61.1%–65.5% (Younesi et al., 2021). In another study, Trichoderma asperellum demonstrated the greatest degree of inhibition of mycelial growth in Fusarium, by 73.29% (Mishra et al., 2022). In another study, seeds treated with T. harzianum and T. viride revealed significant reduction in root rot and wilt disease caused by Rhizoctonia solani and F. oxysporum, under pot conditions (Rudresh, Shivaprakash & Prasad, 2005; Dubey et al., 2012). Nagamani, Biswas & Bhagat (2015) reported biocontrol ability of Trichoderma in controlling F. oxysporum under in vitro conditions and observed 81.1% efficiency of Trichoderma. Similarly, root rot and wilt disease caused by R. solani and F. oxysporum can be successfully controlled by the application of various strains of Trichoderma (Rudresh, Shivaprakash & Prasad, 2005). The introduction of T. harzianum propagules into the soil resulted in an increase of shoot length by 45%, under pot conditions (Yedidia et al., 2001). Similarly, T. harzianum enhanced the fresh shoot weight in chickpea plants by 27.3%, as compared to positive control, in pot experiment (Siddiqui & Singh, 2004).

T. harzianum and Trichoderma atroviride are some species that have multiple interactions with plants and fungal pathogens (Woo et al., 2006). Furthermore, Trichoderma when develop symbiotic interaction with soil microorganisms, boost the organic matter decomposition and release more nutrients that are easily available to plants to achieve sustainable agriculture by enhancing growth, productivity, and yield of crops (Ruiz-Cisneros et al., 2018; Sharma & Sharma, 2020; Turkan et al., 2023). Different species of Trichoderma are known for their ability to control plant diseases, enhancement of the plant growth by modifying the rhizosphere and also by activating the defense mechanisms in plants (Chaverri, Gazis & Samuels, 2011; Keswani et al., 2016). Biocontrol efficacy of Trichoderma depends upon several abiotic conditions, such as soil pH, water holding capacity, and soil and plant root zone temperature. During myco-parasitic mechanism, Trichoderma species secrete different enzymes like glucanase, chitinase, and protease that devastate the fungal cell wall and reduce the disease severity (Mukhopadhyay & Kumar, 2020).

As there are number of reports regarding bio-efficacy of Trichoderma in controlling plant diseases, but such reports are missing for indigenous Trichoderma spp. against the local pathogen. For an effective biocontrol agent (BCA), it is recommended to search the BCA from indigenous environmental niches of the fungal pathogens, as they could be more effective than exotic BCA (Yazid et al., 2023), and could reduce the risk of non target effects on the ecosystem (De Clercq, Mason & Babendreier, 2011). Although field trials are preferred methods to evaluate the crop behavior, studies suggested that outdoor pot trials are better way to evaluate the crop behavior as compared to pot trials conducted under glasshouse and are therefore, a preferred alternative if field trials are not feasible (Pastor, Palacios & Torres, 2023). Studies conducted under controlled conditions like in vitro or glass house conditions for improving agricultural applications is a central matter of debate, demanding a holistic approach (Nelissen, Moloney & Inzè, 2014). Therefore, the present study was designed to examine the effects of indigenous Trichoderma spp. as biocontrol agent against chickpea Fusarium wilt in multiyear outdoor pot trials, in relation to weather conditions, not reported earlier.

Materials & Methods

Collection of the pathogen, Fusarium oxysporum and Trichoderma spp.

Chickpea growing fields in District Chakwal, Punjab, Pakistan were surveyed to find Fusarium wilted chickpea plants and collection of soil samples for the isolation of indigenous Trichoderma spp. Three cm of the upper soil was discarded and five sub-samples were taken randomly at a depth of 20 cm and put into sterilized polyethylene bags and were brought to the laboratory for the isolation of indigenous Trichoderma spp. The soil samples from each site were pooled together to give one composite sample for each location (Ru & Di, 2012). Trichoderma spp. were isolated from sampled soil on potato dextrose agar (PDA) by serial dilution technique and the plates were kept at 26 °C for 4 days. The fungal colonies were purified and kept at 26 °C for 7–8 days. The cultures were maintained on PDA slants (Khandelwal et al., 2012). F. oxysporum was isolated from diseased chickpea plant and Trichoderma asperellum was isolated from the rhizosphere of diseased chickpea plant at location 1 (Morat), while Trichoderma harzianum (strain 1) and Trichoderma harzianum (strain 2) were isolated from the soil of location 2 (Dhok Jamal). These locations were ≈ 2.1 km apart from each other.

Media preparation

Glass flasks were washed with tap water and rinsed thrice with distilled water. Peeled potato slices (200 g) were boiled in 500 mL dH2O for 30 min, then filtered through muslin cloth with the extract saved. A total of 20 g dextrose and 20 g agar were added in potato extract and mixed properly to avoid clot formation. Total volume of solution was made up to 1 L by adding distilled water. Media were sterilized for 20 min at 121 °C. Antibiotic streptomycin was added in the media to prevent bacterial growth and the medium was poured in sterilized glass Petri plates.

Isolation of pathogen (Fusarium) from infected chickpea roots

Infected chickpea roots with symptoms of the wilt disease were sectioned (0.5–1.0 cm), washed with tap water, surface sterilized with clorox (NaOCl) for 5 min, rinsed three times with sterilized dH2O and dried on sterilized filter papers. These root pieces were plated at the rate of five pieces/Petri dish on the PDA medium supplemented with chloramphenicol (0.05 g/L) and incubated at 26 °C for 7 days. The isolated fungi were further sub-cultured on PDA. Pure isolates were observed for their growth patterns and pigmentation on the adverse side of the agar plates. Further microscopic examination was carried out for mycelia and conidial structures using pure culture of F. oxysporum. Pure cultures of the isolated fungi were transferred to PDA slants and stored in a refrigerator at 4 °C. Temporary slide mounts were prepared to confirm identity (Mohamed & Haggag, 2006).

Isolation of Trichoderma spp. from soil through serial dilution method

Stock solution of sample was prepared by dissolving 1 g of soil sample into 10 mL of dH2O. Serial dilutions of samples were prepared at 10−1, 10−2, 10−3 up to 10−7. A total of 100 microliters of 10−3 of the prepared dilution was spread uniformly on the PDA in a Petri dish with the help of a glass spreader. Plates were covered with parafilm to avoid microbial contamination and placed in an incubator at 26 °C for 3 days. After 3 days, fungal growth was observed on PDA plates (Gil, Pastor & March, 2009).

Purification of fungal spp.

Different fungal cultures were taken from actively growing ends with inoculating needle. Then these fungal cultures were inoculated on to fresh media plates. Petri plates were closed with parafilm and labeled. Plates were incubated in growth chamber and the fungal growth was checked over time.

Identification of Trichoderma spp. and F. oxysporum on morphological basis

Pure cultures of Trichoderma isolates and F. oxysporum were observed for morphological traits e.g., shape and color of colony, spore size, shape and color of conidia and conidiophores, as well as structure and branching of mycelia, based on morphological descriptions (Watanabe, 2010).

Molecular identification of fungal species genomic DNA extraction and PCR of ITS of rDNA

DNA extraction and purification was done by following the procedure as described by (Lacap, Hyde & Liew, 2003; Ghosh & Pal, 2017). The internal transcribed spacer (ITS) regions were amplified by PCR using DNA amplification reagent kit manual (GeNei) with fungal specific forward primer ITS-1 F (Gardes & Bruns, 1993) and the reverse primer ITS-4 (White et al., 1990). PCR was done by adopting the protocol as described by Ghosh & Pal (2017), modified from (Gardes & Bruns, 1993). Amplified products were analyzed on 1% agarose gel containing 12 µL of ethidium bromide in 0.5 X Tris-borate EDTA (TBE) buffer. The purified PCR products were sequenced and obtained sequences were subjected to BLAST (https://www.ncbi.nlm.nih.gov/), and were compared with fungal sequences available at NCBI. The sequences were further manually edited and aligned using CLUSTAL W program in MEGA 11 for phylogenetic analysis and to study the diversity among tested isolates.

Dual culture experiments

The Trichoderma spp. were initially assessed for their antagonistic activity against F. oxysporum by in vitro dual culture technique. A 5 mm diameter mycelial disc from the margins of the 7 days old culture of Trichoderma spp. and the F. oxysporum were placed on the opposite of the plate at equal distance. The Petri plates for each test isolate were arranged in a completely randomized design (CRD) with three replicas and incubated at 25 ± 1 °C for 7 days. Radial growth was measured after 7 days of the incubation period and % age inhibition was determined as follows;

L = [(C − T)/C] × 100

where L = inhibition of mycelial growth; C = growth measurement of the pathogen in control and T = growth of the pathogen in the presence of Trichoderma (Wonglom et al., 2019).

Mass culturing of Fusarium oxysporum and Trichoderma spp.

For mass culturing of F. oxysporum, sorghum seeds were autoclaved in heat resistant polythene bags and were inoculated with spores of F. oxysporum, under aseptic conditions, in a laminar air flow cabinet. These bags were incubated at 26 ± 2 °C for ten days until all the grains were fully covered with fungal spores/mycelia. After 10 days, grains were air dried and ground to powder.

Preparation of carrier based Trichoderma formulations and Fusarium inocula

Mass production of Trichoderma spp. included talc based formulation. In talc based formulation, PDA was used for initial growth of Trichoderma spp. in Petri plates. A total of 1,000 mL PDA was prepared in conical flask. Flask was plugged with cotton and covered with aluminum foil to avoid air-borne contamination & sterilized by autoclave at 15 lbs (121 °C temperature) for 15 min. PDA medium was poured in Petri dishes and allowed to solidify. Trichoderma spores were transferred in Petri plates and incubated at 25 ± 2 °C for 6 days. When Trichoderma covered the whole plate, then spore suspension was prepared. One mL of dH2O was added in Trichoderma culture with the help of a pipette. Trichoderma spores were suspended in 100 mL distilled water. After washing of chickpea seeds, these seeds were soaked in 2% sucrose solution for 6 hrs (Batta, 2004). Then extra sucrose solution was removed from chickpea seeds. These seeds were packed within polypropylene bags and sealed. PP-bags were autoclaved at 15 lbs (121 °C) for 15 min. After sterilization, one mL spore suspension was added on autoclaved chickpea seeds and incubated at 25 ± 2 °C for 15 days. After fifteen days, Trichoderma impregnated chickpea seeds were ground in a mixer. Fusarium inoculum was prepared in the same manner as described for Trichoderma, except that Fusarium mass multiplication was made on sorghum seeds (Shanmugam, Chugh & Sharma, 2015).

A total of 2 gm culture of each isolate was mixed well with 100 mL dH2O and filtered through muslin cloth. Few drops of Tween 20 (Alkest TW 20) were added in the spore suspension as a wetting agent (Jamil et al., 2010). Spore concentration was measured/gram of chickpeas using hemocytometer. Then the required quantities of chickpeas with Trichoderma inoculum were added into 1 kg sterilized talc [Mg3Si4O10(OH)2] (Batch # S.15442; Rurka Export China Food and Pharma Grade Osmanthus Brand), and 5 gm of carboxymethyl cellulose was applied to seeds and mixed thoroughly to get fine coating on seeds. The treated seeds were spread over blotter paper and air dried (Pandey, Gohel & Jaisani, 2017). The product was packed in PP-bags and sealed. The final colony forming units (CFU) in all Trichoderma spp. were adjusted at 1.5 × 107 CFU/g of product, while, the final CFU in F. oxysporum was adjusted at 1.2 × 105 CFU/g of the product. Fine coated seeds were sown in each sterilized pot (Bhagat & Pan, 2011).

Pot experiments

Pot experiments were performed to check the efficacy of Trichoderma as biocontrol agent against Fusarium wilt. Chickpea susceptible variety (Pb-91) (Rashid et al., 2013) was chosen as a test crop. Certified seeds of chickpea variety were purchased from the local market.

Preparation and sterilization of soil mixture

Silty loam soil collected from a field at National Agricultural Research Centre (NARC) Islamabad, Pakistan was passed through a sieve. The soil and river sand were mixed in a ratio of 4:1 and filled in sacks and autoclaved. Sterilized earthen pots were filled with 1.5 kg of soil (Siddiqui & Singh, 2004).

Raising and maintenance of chickpea plants in pots

Chickpea seeds were disinfected with 0.1% mercuric chloride and rinsed thrice with autoclaved water. Five seeds were sown (during the month of November each year) into each pot at equal distance from each other. Thinning was carried out after successful emergence of seedlings to three chickpea plants per pot.

Following treatments were investigated in pot experiments;

T1 = (Non-infested control) without Trichoderma and Fusarium

T2 = (Infested control) Fusarium (1.2 × 105 CFU/g of product), 10 g

T3 = Fusarium (1.2 ×105 CFU/g of product), 10 g + Trichoderma asperellum (1.5 × 107 CFU/g of product) 8 g (concentration 1)

T4 = Fusarium (1.2 × 105 CFU/g of product), 10 g + Trichoderma asperellum (1.5 × 107 CFU/g of product) 12 g (concentration 2)

T5 = Fusarium (1.2 ×105 CFU/g of product), 10 g + Trichoderma harzianum (strain 1) (1.5 × 107 CFU/g of product) 8 g (concentration 1)

T6 = Fusarium (1.2 × 105 CFU/g of product), 10 g + T. harzianum (strain 1) (1.5 × 107 CFU/g of product) 12 g (concentration 2)

T7 = Fusarium (1.2 × 105 CFU/g of product), 10 g + Trichoderma harzianum (strain 2) (1.5 × 107 CFU/g of product) 8 g (concentration 1)

T8 = Fusarium (1.2 × 105 CFU/g of product), 10 g + T. harzianum (strain 2) (1.5 × 107 CFU/g of product) 12 g (concentration 2)

T9 = Fusarium (1.2 × 105 CFU/g of product), 10 g + T. asperellum, T. harzianum (strain 1), T. harzianum (strain 2) consortium (1.5 × 107 CFU/g of product) 2.66 g of product of each Trichoderma (concentration 1)

T10 = Fusarium (1.2 × 105 CFU/g of product), 10 g + T. asperellum, T. harzianum (strain 1), T. harzianum (strain 2) consortium (1.5 × 107 CFU/g of product) 4 g of product of each Trichoderma (concentration 2)

Experimental design was CRD. Pot experiments were executed in two consecutive seasons (2020–2021 & 2021–2022). Each treatment was replicated three times. Field related experiments were approved by Crop Diseases Research Institute (CDRI), National Agricultural Research Centre, Islamabad, vide Dy No. 10712.

Disease measurements, disease incidence and severity

In case of disease measurements in pot experiments, disease incidence (DI) and disease severity (DS) were determined (Khan, Khan & Mohiddin, 2004; Nandeesha & Huilgol, 2021).

Following formula was used for determination of DI and DS. For DS, 0–5 scale was adopted; 0 = no wilt, 1 = 1–20%, 2 = 21–40%, 3 = 41–60%, 4 = 61–80%, & 5 = 81–100% (Khan, Khan & Mohiddin, 2004; Jamil & Ashraf, 2020). These data in pots were recorded 68 days after sowing (DAS), in the years 1 & 2.

Wilt incidence (%) = TotalnumberofwiltedplantsTotalnumberofplantsobserved×100

Disease severity = Number of branches, twigs, or leaves showing wilt symptoms/total number of branches, twigs, or leaves.

Harvesting and data collection in pot experiments

Pot data were taken after 101 DAS, during the month of February each year. Following morphological parameters were recorded in pot experiments; shoot length, shoot fresh and dry weights. In case of disease measurements, DI and DS were determined (Dubey, Suresh & Singh, 2007).

Soil analyses

Soil analyses of chickpea growing fields used to isolate Fusarium and indigenous Trichoderma spp. and of pot soils were carried out by following the procedures as described by Estefan, Sommer & Ryan (2013) (Table 1).

Table 1 Soil composition of chickpea growing fields used to isolate Fusarium and indigenous Trichoderma spp. and of pot soils.

Treatments	Soil texture	pH	SOM	NO3-N	Available P	Available K	Zn	Fe	Cu	
Location 1	Sandy loam	7.50 ± 0.14b	0.48 ± 0.04c	3.5 ± 0.1a	5.2 ± 0.35b	92 ± 3.6b	0.91 ± 0.03a	6.03 ± 0.2c	1.21 ± 0.04a	
Location 2	Sandy loam	7.9 ± 0.4a	0.64 ± 0.03a	2.1 ± 0.1d	6.5 ± 0.37a	101 ± 2.8a	0.94 ± 0.07a	5 ± 0.1d	1.03 ± 0.06b	
Pot year 1	Sandy loam	7.4 ± 0.11b	0.55 ± 0.02b	2.65 ± 0.05b	6.14 ± 0.2a	79.3 ± 1.5c	0.66 ± 0.03b	9.7 ± 0.7a	0.37 ± 0.01d	
Pot year 2	Sandy loam	7.51 ± 0.09b	0.6 ± 0.03ab	2.33 ± 0.06c	5.6 ± 0.1b	80 ± 1.3c	0.62 ± 0.02b	8.8 ± 0.3b	0.44 ± 0.02c	
Notes.

Values indicate averages of three repetitions ± standard deviation. In each column, averages with common alphabets do not differ at P = 5%, as computed by Fisher’s LSD test using Minitab 20.2. Significance was determined within each column.

SOM = soil organic matter%; Nitrate-nitrogen (NO3-N) mg/kg; Available phosphorus (P) mg/kg; Available potassium (K) mg/kg; Zinc (Zn) mg/kg; Iron (Fe) mg/kg; Copper (Cu) mg/kg

Environmental data

Environmental data were collected from the Pakistan Meteorological Department, Islamabad, Pakistan (Table 2).

Table 2 Average rainfall (mm) and temperature (degree celsius) data.

Rainfall	
Years	Nov	Dec	Jan	Feb	
20–21	92.83	21.21	11.73	8.02	
21–22	0.01	9.82	174.52	19.22	
Max-Temperature	
Years	Nov	Dec	Jan	Feb	
20–21	23.1	18.9	19.2	24	
21–22	25.2	19.9	16.3	20	
Min-Temperature	
Years	Nov	Dec	Jan	Feb	
20–21	7.1	4.2	2.6	6.8	
21–22	6.9	2.9	4.7	6.6	

Confirmation of Koch’s postulates

To fulfill Koch’s postulates, healthy chickpea plants were inoculated with F. oxysporum grown on a culture that had been isolated from Fusarium wilt affected chickpea root. Then the plant was studied for disease symptoms and its comparison to the infected plants that had been used for the isolation of F. oxysporum.

Statistical analyses

All the data were analyzed by ANOVA, and after performing ANOVA, Fisher’s LSD test was performed at P = 5%, using Minitab 20.2.

Results

Morphological identification of isolated Fusarium and Trichoderma strains

F. oxysporum showed two types of conidia; macroconidia were boat shaped and four celled, while microconidia were ellipsoidal shaped and one celled. Chlamydospores were brown in color and globose shaped. Colony was white in color. Microconidia were with size of 6.1 µm × 3.2 µm. While, macroconidia were sickle shaped with an average size of 30.7 µm × 3.4 µm.

T. harzianum showed conidiophores and hyaline phialides were short and thick. Conidia were ovate shaped and one-celled. Conidiophore size was 78 µm, while conidia were 2.4 µm in diameter. Colony was dark green in color on growth medium. On the other hand, T. asperellum had slightly ovoidal shaped conidia, having size of 2.91 µm × 2.37 µm. Conidia were one-celled.

Phylogenetic analysis of Trichoderma spp.

Phylogenetic analysis had shown that the three isolates of the Pakistani Trichoderma collected for this study clustered in three clades with reference sequences from different regions of the world. The Trichoderma sp. (SA623043) reported in this study clustered with a sequence of Trichoderma from Pakistan, China, and Africa with 99% bootstrap value. The Trichoderma sp. (SA1522759) reported in this study clustered with two sequences of Trichoderma sp. from Nigeria and Brazil that was part of another clade with sequences from Brazil and Nigeria. The third isolate of Trichoderma sp. (W116499) reported in this study clustered with number of sequences of Trichoderma from Pakistan, China, Portugal, and Taiwan with 99% bootstrap value. The phylogenetic framework based on ITS sequences clarified the T. harzianum, although belonging to the same species, yet they phylogenetically belong to different clades (Fig. 1).

Figure 1 Phylogenetic tree of Trichoderma isolates.

Phylogenetic analysis of F. oxysporum f. sp. ciceris

Phylogenetic analysis had shown that the isolate of the Pakistani F. oxysporum f. sp. ciceris analysed in the present study grouped in the major clade with reference sequences from Pakistan, India, Egypt, Turkey, and China with 48% bootstrap value. The F. oxysporum f. sp. ciceris (OR808009) reported in this study clustered in a sub-clade with a sequence of Fusarium from India and Egypt.

The evolutionary history was deduced with Neighbor-Joining technique (Saitou & Nei, 1987). The % age of replica trees in which the associated taxa clustered together in the bootstrap test (500 replicas) are shown next to the branches (Felsenstein, 1985). The evolutionary distances were computed by the Maximum Composite Likelihood technique (Tamura, Nei & Kumar, 2004) and are in the units of number of base substitutions per site. This analysis involved 17 nucleotide sequences including nucleotide sequence (OR808009) reported in present study. All obscure positions were removed for each sequence pair (pairwise deletion option). There were a total of 1,244 positions in the final dataset. Evolutionary analyses were performed on MEGA 11 (Tamura, Stecher & Kumar, 2021) (Fig. 2).

Figure 2 Phylogenetic tree of Fusarium oxysporum f. sp. ciceris.

Effect of various Trichoderma spp. on Fusarium in dual culture experiments

In dual culture experiments, T. asperellum inhibited the F. oxysporum f. sp. ciceris by 37.6%, while, T. harzianum strains 1 and 2 significantly inhibited the radial/mycelial growth of F. oxysporum by 40% and 42%, respectively (Table 3).

Table 3 Percentage inhibition of F. oxysporum f. sp. ciceris in dual culture experiments.

Serial no.	Treatments	Measurements of colonies in centimetres	% age inhibition of Fusarium	
		Length	Width	Average		
1-	Fusarium in control	4.4 ± 0.79a	6.5 ± 0.76a	5.45 ± 0.62a	–	
2-	Fusarium with Trichoderma asperellum	2.3 ± 0.50b	4.5 ± 0.53b	3.4 ± 0.33b	37.6	
3-	Fusarium with Trichoderma harzianum I	2.4 ± 0.40b	4.1 ± 0.40b	3.25 ± 0.20b	40	
4-	Fusarium with Trichoderma harzianum II	2 ± 0.46b	4.3 ± 0.46b	3.15 ± 0.46b	42	
Notes.

Values are means of 3 replicates ± standard deviation. Standard deviation values were rounded off to 2nd decimal. Values sharing same letter do not differ at P = 5%, as computed by Fisher’s LSD test using Minitab 20.2. Fusarium stands for F. oxysporum f. sp. ciceris. Significance was determined within each column

Effects on shoot length, shoot fresh weight, and shoot dry weight of chickpea plant in pot trials (year 1)

The Fusarium infested chickpea plants (infested control, IC) showed a 27% decrease in shoot length as compared to non-infested control (NIC). Treatments with T. asperellum (T.a) concentration 1 and concentration 2, T. harzianum strain 1 (Th S1), concentration 1 and concentration 2, T. harzianum strain 2 (Th S2) concentration 1 and concentration 2, and consortium of all these Trichoderma species (concentration 1&2), significantly increased the shoot length of chickpea plants by 20% and 31%, 32% and 39%, 39% and 46%, 56% and 69%, respectively, over IC. Moreover, consortium of all these Trichoderma species at concentrations 1 and 2, significantly enhanced the shoot length of chickpea plants by 13% and 23%, respectively, in comparison with NIC (Fig. 3A).

Figure 3 Effect of treatments on shoot length (A), shoot fresh weight (B), shoot dry weight (C) of chickpea in pot trials (Year 1).

Bars with common alphabets do not differ at P = 5%, as computed by Fisher’s LSD test using Minitab 20.2. Y-error bars depict the standard error of three repetitions. NIC = Non infested control; IC = Infested control; T.a = Trichoderma asperellum, T.h S1 = Trichoderma harzianum strain 1; T.h S2 = Trichoderma harzianum strain 2; Cons = consortium of T.a, T.h S1 & T.h S2; conc. = concentration.

The chickpea plants that were infested with Fusarium experienced a 43% decrease in the shoot fresh weight as compared to NIC. However, all treatments involving Trichoderma showed a significant increase in shoot fresh weight, in comparison with IC, effectively counteracting the negative effects of the F. oxysporum pathogen. The effectiveness of Trichoderma in controlling F. oxysporum was found to depend on the dosage. Treatments with T.a, Th S1, Th S2, and a combination of these Trichoderma species, all significantly increased the shoot fresh weight of chickpea plants. These increases were 21% and 32%, 31% and 37%, 38% and 44%, and 60% and 67% at concentrations 1 and 2, respectively, compared to the IC (Fig. 3B).

IC of chickpea plants, affected by Fusarium, exhibited a 41% decrease in shoot dry weight, when compared to the NIC. However, all treatments involving Trichoderma showed a significant increase in shoot dry weight, effectively mitigating the negative effects of the pathogen, F. oxysporum. The efficacy of Trichoderma in controlling F. oxysporum was found to be dependent on the dosage. Treatments with T.a, Th S1, Th S2, and a consortium of these Trichoderma species significantly increased the shoot dry weight of chickpea plants. The increases were 22% and 32%, 32% and 38%, 37% and 42%, and 56% and 68% at concentrations 1 and 2, respectively, compared to the IC. Furthermore, the consortium of all these Trichoderma species at concentrations 1 and 2 showed non-significant differences when compared to the NIC (Fig. 3C).

Effect on disease incidence and disease severity of chickpea plants in pot trials (year 1)

There was 88.9% disease incidence (DI) in Fusarium infested chickpea plants, as compared to NIC. All the treatments with Trichoderma demonstrated a significant decrease in DI of chickpea plants. Treatments with T.a, Th S1, Th S2, and consortium of all these Trichoderma spp. significantly decreased the DI of chickpea plants by 77.8% and 66.7%, 66.7%, 55.6% and 44.4%, 33.3%, 33.3% and 22.2%, at concentrations 1 and 2, respectively, over IC. Moreover, consortium of all these Trichoderma spp. at concentrations 1 and 2, significantly reduced the DI on chickpea plants by 33.3% and 22.2%, respectively, in comparison with infested control (IC).

In year 1, IC had a disease severity (DS) of 4.7, whereas, there were no disease symptoms in NIC. Moreover, all the treatments using Trichoderma showed significant reduction on DS in the chickpea plants and this reduction in DS was dose dependent. Treatments using T.a, Th S1, Th S2, and a combination of all these Trichoderma, all significantly decreased the DS in the chickpea plants. The reductions in DS were 21% and 23%, 29%, 31%, and 40%, and 47%, 71%, and 86% at concentration 1 and concentration 2, respectively, compared to the IC (Table 4).

Table 4 Disease incidence (DI) and disease severity (DS) on chickpea plants in pots (year 1 and year 2).

Treatments	Year 1 (2020–2021)	Year 2 (2021–2022)	
	DI	DS	DI		DS	
		Reduction over IC (%)		Reduction over IC (%)		Reduction over IC (%)		Reduction over IC (%)	
NIC	0 ± 0e	–	0 ± 0e	–	0 ± 0f	–	0 ± 0f	–	
IC	88.89 ± 19.2a	0	4.67 ± 0.33a	0	77.78 ± 19.3a	0	4.11 ± 0.84a	0	
T.a conc.1	77.78 ± 19.2ab	12.5	3.68 ± 0.86ab	21.2	66.67 ± 0ab	14.3	3.55 ± 0.38ab	13.6	
T.a conc.2	66.67 ± 33.3a-c	25	3.58 ± 0.82ab	23.3	55.56 ± 19.2a-c	28.6	2.89 ± 0.19b	29.7	
T.h S1 conc.1	66.67 ± 33.3a-c	25	3.33 ± 0.85b	28.7	55.56 ± 19.2a-c	28.6	2.89 ± 0.19b	29.7	
T.h S1 conc.2	55.56 ± 19.2a-d	37.5	3.23 ± 0.68b	30.8	44.44 ± 19.2b-d	42.9	2.67 ± 0.57bc	35	
T.h S2 conc.1	44.44 ± 19.2b-d	50	2.8 ± 0.17b	40	44.44 ± 19.2b-d	42.9	2.67 ± 0.57bc	35	
T.h S2 conc.2	33.33 ± 0c-e	62.5	2.47 ± 0.4bc	47.1	33.33 ± 0c-e	57.1	1.78 ± 0.38cd	56.7	
Cons conc.1	33.33 ± 33.3c-e	62.5	1.34 ± 1.35cd	71.3	22.22 ± 19.2d-f	71.4	1.33 ± 1.15d	67.6	
Cons conc.2	22.22 ± 19.2de	75	0.67 ± 0.57de	85.7	11.11 ± 19.2ef	85.7	0.33 ± 0.57e	92	
Notes.

NIC Non infested control

IC Infested control

T.a Trichoderma asperellum

T.h S1 Trichoderma harzianum strain 1

T.h S2 Trichoderma harzianum strain 2

Cons consortium of T.a, T.h S1 & T.h S2

conc. concentration

Values indicate averages of three repetitions ± standard deviation. In each column, averages with common alphabets do not differ at P = 5%, as computed by Fisher’s LSD test using Minitab 20.2. Significance was determined within each column.

Effect on disease incidence and disease severity of chickpea plants in pot trials (year 2)

There was 77.8% DI in Fusarium infested chickpea plants (IC), as compared to NIC. All the treatments with Trichoderma demonstrated a significant decrease in DI of chickpea plants. The increase in the concentration of Trichoderma resulted in a decrease of DI. Treatments with T.a, Th S1, Th S2, and consortium of all these Trichoderma species significantly decreased the DI of chickpea plants by 66.7% and 55.6%, 55.6%, 44.4% and 44.4%, 33.3%, 22.2% and 11.1%, at concentrations 1 and 2, respectively, over IC. Moreover, consortium of all these Trichoderma spp. at concentrations 1 and 2, significantly reduced the DI of chickpea plants by 22.2% and 11.1%, respectively, in comparison with infested control (IC).

In year 2, IC revealed DS score of 4.1. Trichoderma showed a significant decrease in DS caused by the pathogen in the chickpea plants. Treatments using T.a, Th S1, Th S2, and a combination of all these Trichoderma species significantly reduced the DS in the chickpea plants. The reductions in DS were 14% and 30%, 30%, 35%, and 35% at concentration 1, and 57%, 68%, and 92%, at concentrations 1 and 2, respectively, in comparison with IC (Table 4).

Effects on shoot length, shoot fresh weight, and shoot dry weight of chickpea plants in pot trials (year 2)

In year 2 pot experiments, infested control showed a 23% decrease in shoot length as compared to NIC. All the treatments with Trichoderma demonstrated a significant increase in shoot length of chickpea plants, minimizing the negative effects of the pathogen, F. oxysporum. The effect of Trichoderma in controlling the F. oxysporum was found dose dependent. Treatments with T.a, Th S1, Th S2, and consortium of all these Trichoderma species significantly increased the shoot length of chickpea plants by 26% and 36%, 44% and 47%, 48% and 54%, 64% and 72%, at concentrations 1 and 2, respectively, over IC. Moreover, consortium of all these Trichoderma species at concentrations 1 and 2, significantly enhanced the shoot length of chickpea plants by 27% and 33%, respectively, in comparison with NIC (Fig. 4A).

Figure 4 Effect of treatments on shoot length (A), shoot fresh weight (B), & shoot dry weight (C) of chickpea in pot trials (Year 2).

Bars with common alphabets do not differ at P = 5%, as computed by Fisher’s LSD test using Minitab 20.2. Y-error bars depict the standard error of three repetitions. NIC = Non infested control; IC = Infested control; T.a = Trichoderma asperellum, T.h S1 = Trichoderma harzianum strain 1; T.h S2 = Trichoderma harzianum strain 2; Cons = consortium of T.a, T.h S1 & T.h S2; conc. = concentration.

The Fusarium infested chickpea plants revealed 38% decline in the shoot fresh weight compared to NIC. All treatments involving Trichoderma either alone or in the form of consortia, showed significant enhancement in shoot fresh weight of chickpea plants, indicting the beneficial effect of Trichoderma against F. oxysporum. Treatments with T.a, Th S1, Th S2, and a combination of these Trichoderma species, all significantly increased the shoot fresh weight of chickpea plants. The increases were 28% and 37%, 45% and 50%, 50% and 57%, and 67% and 73% at concentrations 1 and 2, respectively, compared to the IC. Furthermore, the combination of all these Trichoderma species at concentrations 1, 2 and Th S2 showed non-significant differences, in comparison with NIC (Fig. 4B).

Fusarium infection exhibited a 36% decrease in shoot dry weight of chickpea plants as compared to NIC. Treatments with T.a, Th S1, Th S2, and a consortium of these Trichoderma species significantly increased the shoot dry weight of chickpea plants by 25% and 36%, 44% and 49%, 56% and 63%, and 68% and 75% at concentrations 1 and 2, respectively, compared to the IC (Fig. 4C).

Soil analyses

Soil texture was sandy loam at chickpea growing locations as well as the soil used in pot experiments. There was significant difference in the soil pH and soil organic matter (SOM) of both locations, while the pH and SOM of soil used in pot experiments in both years were non-significant, when compared with each other. NO3-nitrogen, available phosphorus, available potassium, iron, and copper were significantly different at both locations, except zinc. On the other hand, NO3-nitrogen, available phosphorus, iron, and copper were significantly different in the soils used in pots of year 1 and year 2, except available potassium and zinc. Moreover, variable results were recorded when we compare the soil properties used in pot experiments as compared with soil properties of location 1 and 2 (Table 1).

Environmental data

Environmental data for the years 2020–2021 and 2021–2022 showed that during the critical stage (flowering stage susceptible to Fusarium wilt = January), there was high rainfall (Table 2).

Confirmation of Koch’s postulates

The healthy chickpea plants inoculated with F. oxysporum grown on a culture that had been isolated from Fusarium wilt affected chickpea root in the pot trials, developed the same symptoms as inoculated initially.

Discussion

In the present study, Trichoderma species significantly inhibited the mycelial growth of F. oxysporum f. sp. ciceris (FOC) in dual culture experiments. These results are consistent with the findings of Al-Surhanee (2022) who reported the 25% reduction in F. oxysporum in dual culture experiments. Similarly, Trichoderma strain ThrAN-5 showed 69% inhibition in mycelial growth of F. oxysporum (Bhagat & Pan, 2011). Moreover, T. harzianum revealed the in vitro bio-efficacy against different pathogens and exhibited 81% inhibition (Kumari et al., 2020). In another investigation, the Trichoderma isolates exhibited high competitive ability and the synthesized metabolites demonstrated inhibitory effects on the mycelial growth of Fusarium. Trichoderma isolates’ mechanisms of action is primarily linked to the generation of volatile organic compounds. Although Trichoderma exhibited an antagonistic effect in vitro, they did not demonstrate efficacy in controlling Fusarium or promoting chickpea growth under in vivo conditions (Queirozazevedo et al., 2020). But there are numerous studies that reveal the effectiveness of Trichoderma in controlling the pathogens and boosting the plant growth, under pot conditions.

In the present investigation, Trichoderma either alone or in consortia significantly declined the DI and DS in chickpea plants under multiyear pot trials. Our findings are in accordance with the results of Ramanagouda, Naik & Sharma (2022) who reported the effectiveness of T. harzianum against Fusarium wilt of chickpeas caused by FOC. In another investigation, the application of T. harzianum significantly reduced the DI in chickpea plants by 44%, as compared to untreated control (Nandeesha & Huilgol, 2021). The implementation of T. harzianum treatment resulted in significant enhancements in various plant growth parameters of chickpea plants as there was 3% increase in the plant height (Martínez-Martínez et al., 2020). In another pot experiment, T. harzianum increased the shoot length, shoot fresh and dry weight of chickpea plants by 46.9%, 27.3%, and 79.4%, respectively, as compared to Fusarium infested control (Siddiqui & Singh, 2004). In another investigation, T. harzianum increased the shoot height, fresh and dry weight of chickpea plants by 36.5%, 25%, and 33%, respectively, as compared to infested control pots (Meher, Singh & Sonkar, 2018). In another study, there was a significant rise in the dry weight of chickpea by 80% by inoculation of T. harzianum (Yedidia et al., 2001).

Trichoderma species are fast-growing fungi (Dutta et al., 2023), and can colonize the plant root system (Tyśkiewicz et al., 2022) and its surrounding soil (Brotman et al., 2013), competing with Fusarium spp. for space and nutrients (Oszust, Cybulska & Frąc, 2020). These are mycoparasites, attacking and parasitizing other fungi, including Fusarium, and produce the enzymes and secondary metabolites that break down the Fusarium cell walls and cause leakage of cellular contents (Segaran, Shankar & Sathiavelu, 2022). Trichoderma can induce resistance against Fusarium wilt (Ponsankar et al., 2023), by inhibiting the mycelial growth of fungi and by reducing the disease intensity (Sánchez-Montesinos et al., 2021). Some Trichoderma strains produce volatile compounds that inhibit Fusarium growth and directly suppress the Fusarium populations in the soil (Zhang et al., 2021). Trichoderma can enhance nutrient uptake, stress tolerance, and help the chickpea plants to withstand under Fusarium wilt stress and recover more effectively from infection (Singh & Vyas, 2023). In an investigation, T. harzianum significantly reduced the DI by 62%, as compared to positive (pathogen infested) control (Nandeesha & Huilgol, 2021). Similarly, T. harzianum reduced the DI by 75%, as compared to positive control (Ghosh, Banerjee & Sengupta, 2017). Other studies also reported decline in the DI by 65% and 80%, as compared to positive control, by the inoculation of T. harzianum (Ali & Terefe, 2021; Anwar et al., 2022). The findings of another study indicated that the application of Trichoderma viride to chickpea seeds via soil treatment resulted in the lowest incidence of wilt at 21.50%. In an investigation, variations in the effectiveness of Trichoderma species were recorded as the consortium of Trichoderma virens and T. harzianum showed DI by 29.4%. The consortium of T. virens and T. viride showed DI by 31.3%. While, the consortium of T. viride and T. harzianum reduced the DI by 54.9% (Tomar, Thakur & Yadav, 2022). On the other hand, the consortium of two strains of T. harzianum decreased the DI by 128% in tomato plants, in comparison to control (Singh et al., 2014).

In the present study, Trichoderma consortia significantly improved the growth of chickpea plants in pot trials. Similar findings were previously documented by Trivedi et al. (2020) when plants were treated with T. harzianum and reported 29% increase in shoot length, as compared to positive control. Moreover, a consortium of two T. harzianum significantly enhanced the tomato shoot length by 35%, as compared to negative control (Singh et al., 2014). Similarly, T. harzianum increased dry shoot weight by 29.7%, as compared to positive control (Khan et al., 2014). Likewise, T. harzianum increased dry shoot weight by 32.5%, as compared to positive control (Khan, Khan & Mohiddin, 2004). In another in vivo investigation, Trichoderma cerinum caused 13.3% and 58.3% increase in dry weight of chickpea with or without F. oxysporum, respectively (Khare, Kumar & Kim, 2018).

Trichoderma species produce different kinds of compounds and siderophores that boost the organic matter decomposition in the soil (Kubheka & Ziena, 2022). These compounds improve the solubilization of soil nutrients especially phosphorus and iron that are very important nutrients for plant growth and developments (Silva et al., 2023). On the other hand, Trichoderma spp. are also used as biofertilizers due to its nitrogen fixing ability in the root zone that also promote the growth of chickpea and increase the yield of crop (Bononi et al., 2020). Moreover, it can make protective layer around the roots and produce plant growth promoting substances like auxins and gibberellins that proliferate the root system and increase the root surface area of chickpea that ultimately leads to availability of more water and nutrients and these resources also trigger the plant growth (Song et al., 2023).

In the present study, the consortia of three Trichoderma species was found more effective in alleviating the bad impacts of FOC, thereby enhancing the growth of chickpea. Moreover, these enhancement effects were found dose dependent. Comparable results have been reported in previous studies (Kumari et al., 2020; Worlu et al., 2023).

The variations recorded in the DI, DS and growth of chickpea plants during two growing seasons can be attributed to variations in the environmental factors that greatly influenced the infection levels of FOC on one hand while, affecting the plant growth on the other hand, as discussed in earlier investigations. High moisture contents tend to decrease the Fusarium infection in crops (Shabani & Kumar, 2013). Chakrapani et al. (2023) reported 23–27 °C temperature is favorable for the growth of Fusarium. In another study, in the Fusarium spp. control treatment, DI was low (7.8%) at 22 °C, severe (77% to 82.5%) at 27 °C, and moderate (45%) at 32 °C (Larkin & Fravel, 2002). Stover (1953) reported that Fusarium show optimum growth and survival at 15% saturation as Fusarium is aerobic and its growth can be greatly reduced by maintaining soil in saturated conditions. In another study, 40%, 50%, and 60% soil moisture showed maximum (66.7%, 58.3%, and 50%) reduction in disease incidence in bell pepper (Attri, Sharma & Gupta, 2018). Variations recorded in the disease suppression in the present study can be attributed to changes in the weather conditions during the critical stage of chickpea plants where high rain fall during the month of January 2022 benefitted Trichoderma and chickpea plants on one hand while, it resulted in less disease due to inability of Fusarium to survive at higher moisture.

In the present study, soil used in pot experiments was sandy loam. This soil texture corresponds to soil texture of chickpea growing fields to isolate Fusarium and indigenous Trichoderma spp. Sandy loam soil texture influences the optimum growth of Fusarium oxysporum f. sp. ciceri (Mergewar et al., 2023). Similarly, the soil pH = 7 influences the optimum growth of F. oxysporum f. sp. ciceri (Pempee et al., 2020). Soil macro and micronutrients also influence the Fusarium wilt in chickpea. High level of nitrogen increases the wilt incidence. On the other hand, increase in soil phosphorus tends to decrease the wilt incidence. Although, in year 2 the concentration of phosphorus was significantly low, but this difference may not affect the disease in chickpea because the impact of environmental factors on disease cannot be ignored. Similarly, zinc is found to be the most effective micronutrient in controlling the wilt incidence (Nathawat et al., 2024). In the present study, the concentration of Zn in both years of pot experiments was found non significant, so the level of disease on chickpea plants cannot be attributed to Zn concentration. Low iron concentration tends to decrease the Fusarium wilt disease as Pseudomonas spp. present in the soil produce more salicylic acid under low iron concentration and this salicylic acid is responsible for the development of resistance against F. oxysporum f. sp. ciceri in the host plant as compared to conditions under high iron content in soil where less salicylic acid is produced (Saikia et al., 2005). Although the concentration of Fe in year 1 was significantly low as compared to in year 2 but this decreased Fe concentration cannot be viewed as having impact on lower disease observed in year 2 as the soil was sterilized in pots. On the contrary, high copper content in soil is considered beneficial as it induces resistance in plants against wilt incidence (Graham & Webb, 1991). In our study, the significantly higher concentration of Cu in year 2 may be the cause of low disease on chickpea in year 2, but how much concentrations of macro and micronutrients are required to influence chickpea Fusarium wilt disease need to be investigated in further studies.

As the consortia of Trichoderma spp. significantly reduced the Fusarium wilt in chickpeas, so, based on these results, it is strongly recommended that the secondary metabolites that are produced and showed mycoparasitism should be isolated through bioguided bioassays. Moreover, the effects of Trichoderma species either singly or in the form of consortia, need to be investigated for their effects on plant growth in the absence of pathogen. Similarly, the effects of Trichoderma as well as F. oxysporum in chickpea and other crops need to be investigated on the soil properties and nutritional characteristics, not covered in the present study.

Conclusions

In the present study, formulations containing T. asperellum, T. harzianum strain 1, and T. harzianum strain 2, either singly or in the form of consortia, effectively controlled the Fusarium wilt disease in chickpea in both years of outdoor pot experiments. However, the formulation comprising the combination of these Trichoderma species in the form of consortia was found more effective in controlling the wilt disease in chickpea. This treatment also enhanced the morphological growth of chickpea plants and increased the shoot length, shoot fresh, and dry weight of chickpea plants during the years 1 and 2, respectively, in comparison with infested control plants. Although Trichoderma spp. exhibited consistency in controlling the Fusarium infection in chickpea plants, the variations of weather during the susceptible stage of chickpea plants influenced the disease levels as well as plant growth. The findings of present study suggest the use of Trichoderma spp. to manage the Fusarium wilt of chickpeas.

Supplemental Information

Supplemental Information 1 Author photo gel picture, Fasta Sequence Fusarium oxysporum f. sp. ciceris & Sequences for NCBI Bankit

The gel picture for gel run, Fasta Sequence Fusarium oxysporum f. sp. ciceris and sequences for NCBI Bankit. Photo Credit: Safeer Akbar Chohan

Supplemental Information 2 Raw values for statistical analysis and Minitab results

In pot experiments, Year 1 & 2, all values of disease related parameters and morphological parameters.

Supplemental Information 3 Soil analyses

All Minitab results are given.

Supplemental Information 4 Experiment picture showing the effects of treatments on chickpea

Photo Credit: Safeer Akbar Chohan.

Authors are grateful Prof. Dr. Zia ur Rehman and Mr. Asad Farooq for facilitating in morphological identifications & molecular analysis of fungal isolates.

Additional Information and Declarations

Competing Interests

Author Contributions

Field Study Permissions

DNA Deposition

Data Availability

The authors declare there are no competing interests.

Safeer A. Chohan performed the experiments, prepared figures and/or tables, and approved the final draft.

Muhammad Akbar conceived and designed the experiments, analyzed the data, prepared figures and/or tables, authored or reviewed drafts of the article, and approved the final draft.

Umer Iqbal conceived and designed the experiments, authored or reviewed drafts of the article, and approved the final draft.

The following information was supplied relating to field study approvals (i.e., approving body and any reference numbers):

Field related experiments were approved by Crop Diseases Research Institute (CDRI), National Agricultural Research Centre, Islamabad, vide Dy No. 10712.

The following information was supplied regarding the deposition of DNA sequences:

The sequences described are available at GenBank: Trichoderma asperellum, Isolate W116499, ON862745; Trichoderma harzianum Isolate SA623043, ON862746; Trichoderma harzianum Isolate SA1522759 ON862747; Fusarium oxysporum f. sp. ciceris, Isolate W04FOC, OR808009.

The following information was supplied regarding data availability:

The raw data and the raw measurements are available in the Supplemental File.

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
