# Peer review of "Trichoderma based formulations control the wilt disease of chickpea (Cicer arietinum L.) caused by Fusarium oxysporum f. sp. ciceris, better when inoculated as consortia: findings from pot experiments under field conditions"

_PeerJ, doi:10.7717/peerj.17835_

## Round 0.1 · original submission · Minor Revisions

We received three assessments of your work and all of them are positive. There are concerns though, which need to be attended to, but I think this will not be a problem for the authors. The recommendations include the improvement of figures, the rewriting of some manuscript sections, and the inclusion of more representative figures showing the beneficial effect of Trichoderma.

Reviewer 1 ·

Basic reporting

In general, the manuscript is easy to follow and the objective to be achieved is understood. The approach to conducting the experiments is appropriate. The raw data is shown, except that the data where the BLAST alignment was performed cannot be accessed ((http://blast.ncbi.nlm.nih.gov/Blast).

Experimental design

The methods used are adequately described and with them, the hypothesis raised about the use of Trichoderma as a biocontroller of the Fusarium pathogen can be verified.

Validity of the findings

Although the results show that the consortia are effective in reducing the incidence, as well as the severity of the infection caused by Fusarium. I consider it necessary to access some of the representative images of these experiments, where the beneficial effects of Trichoderma are shown. With their respective controls. These images could be placed in supplementary material.

Additional comments

In the materials and methods section, there are some typographical errors.
for example: ml --> mL; gm --> mg; kgs --> kg

From the results section onwards, you can abbreviate the name of the microorganisms you mentioned before. For example: Fusarium oxysporum --> F. oxysporum. Etc.

Reviewer 2 ·

Basic reporting

In this work, Chohan et al. report the use of Trichoderma as a biocontrol agent to control the wilting of chickpea plants caused by Fusarium. Although several similar studies have already been published, the authors carried out these studies with indigenous strains isolated from the place where the disease was found, a situation that had not been explored.
The paper is generally acceptable; however, there are points in the structure that must be addressed. The antecedents shown in the introduction are repeated in the first part of the discussion. It is necessary to restructure both sections so that the discussion mainly addresses the data generated in the work.

Experimental design

The paper complies with the journal requirements in terms of aims and scope. The project addresses the study of antagonism between Trichoderma and Fusarium with the novelty of doing it with indigenous strains of Trichoderma.

The article mentions that the causal agent of chickpea wilt was isolated from diseased plants in a region of Pakistan and the indigenous strains of Trichoderma from soil samples from this region. However, the area covered by this region and how close the diseased plants were to the soil sample from which the Trichoderma strains were isolated are not specified. This is important to determine the environmental correlation of the strains studied, especially when considering Trichoderma strains as indigenous. The soil composition in this area needs to be reported, and its possible correlation with the soil formulated to carry out the experiments. At the very least, this should be addressed in the discussion of results.

Validity of the findings

The conclusions are presented as a list of results. It is desirable to present them as concrete statements. Also, the perspectives of the work are presented in the conclusions section, which should be part of the discussion section and not the conclusions.

Additional comments

In general, the communication is good work; once the suggested corrections have been completed, I consider the article ready for publication for the next revision in order to be published.

Reviewer 3 ·

Basic reporting

This is good study conducted by the authors. The data presented in this article technically sound in its field. Overall structure of the article is good.

Experimental design

Research design of the study is up to the standards as per pervious methods reported

Validity of the findings

no comments

Additional comments

Fig. 1 resolution is not good, resolution should be like that of fig. 2.
Delete the old reference of 70s or 80,

---

## Round 0.2 · Minor Revisions

The authors addressed all concerns raised by the reviewers, except the following:

"I consider it necessary to access some of the representative images of these experiments, where the beneficial effects of Trichoderma are shown. With their respective controls."

The authors consider this information relevant but mention the quality of the figures is poor. The authors are encouraged to include these figures as supplementary material. Alternatively, new pictures may be prepared by the authors, aiming to improve the quality.

---

## Round 0.3 · accepted · Accept

The authors addressed the Reviewers' concerns and consequently, it is now suitable for publication.